# Inflammatory Surrogate Parameters for Predicting Ifosfamide-Induced Neurotoxicity in Sarcoma Patients

**DOI:** 10.3390/jcm11195798

**Published:** 2022-09-30

**Authors:** Moritz Schmidt, Katrin Benzler, Ulrich M. Lauer, Lars Zender, Clemens Hinterleitner, Martina Hinterleitner

**Affiliations:** 1Department of Medical Oncology & Pneumology (Internal Medicine VIII), University Hospital, 72076 Tuebingen, Germany; 2DFG Cluster of Excellence 2180 ‘Image-Guided and Functionally Instructed Tumor Therapy’ (iFIT), University of Tuebingen, 72076 Tuebingen, Germany; 3Cancer Biology and Genetics, Memorial Sloan Kettering Cancer Center, New York, NY 10013, USA

**Keywords:** sarcoma, ifosfamide, toxicity

## Abstract

Sarcomas compromise a heterogenous group of tumors of a mesenchymal origin. Although treatment options in many solid tumors have evolved over the past decades, the treatment of advanced sarcoma is still based on conventional chemotherapeutic agents. Beside anthracyclines, alkylating agents such as ifosfamide are frequently used in sarcoma treatment. However, treatment with ifosfamide can cause severe dose- and treatment-limiting side effects, such as ifosfamide-induced neurotoxicity (IIN). Especially in sarcoma, consecutive risk assessment analyses investigating the individual factors associated with the increased incidence in IIN, remain insufficient so far. In this retrospective analysis, we investigated 172 sarcoma patients treated with ifosfamide. Out of 172 patients, 49 patients (28.5%) developed IIN. While gender, age, histologic origin, and tumor stage were not associated with the occurrence of IIN, infusion times, simultaneous radiotherapy, and concomitant use of opioids or anticonvulsants affected the risk of developing IIN. Sarcoma patients with IIN showed an alteration in several inflammatory markers, including a lower lymphocyte count, hemoglobin levels, and calcium levels, as well as elevated GGT, sodium, and CRP levels. Remarkably, the occurrence of IIN was associated with a worse prognosis regarding progression free and overall survival. In addition, high CTCAE grades were negatively associated with overall survival in sarcoma. The observation that an inflammatory state is associated with an increased risk of IIN in sarcoma patients can be used prospectively to further investigate the relationship of inflammation and IIN. In addition, the easily accessible blood markers used in our study to predict IIN can be incorporated into clinical decision making.

## 1. Introduction

Despite considerable advances in modern cancer treatment, systemic chemotherapy remains an important component of several multimodal treatment regimens. Although various chemotherapeutic drugs prolong overall (OS) and progression free survival (PFS), chemotherapy-associated toxicities diminish the treatment efficacy significantly.

Sarcoma represent a rare heterogenous group of mesenchymal malignancies with more than 100 different biological entities, and are conventionally distinguished into two superordinate categories—bone sarcomas (BS) and soft tissue sarcomas (STS) [1]. Sarcoma include both slow-growing tumors and highly malignant histological subtypes requiring a multimodal therapy regimen [2,3]. These interdisciplinary treatment concepts include surgery, radiotherapy, and systemic cancer therapy, as well as, in selected cases, regional hyperthermia [3,4,5,6,7,8]. In sarcoma, systemic chemotherapy-based treatment still plays a central role in patients with both curative and palliative disease [4,8]. Regarding the clinical data existing so far, anthracyclines alone or in combination with the alkylating agent ifosfamide are most commonly used [4,7]. Further treatment options such as trabectedin, eribulin, pazopanib, gemcitabine-based combinations, doxorubicin, cisplatin, etoposide, vincristine, and methotrexate depend on the primary histology, patient age, and comorbidities [7,9,10,11,12]. Novel chemotherapy-free approaches including epigenetic regulation via inhibition of the enhancer of zeste homolog 2 (EZH2) with tazemeostat in epithelioid sarcoma, mouse double minute 2 homolog (MDM2) inhibitors, cyclin-dependent kinase (CDK) 4/6 inhibitors or multi-kinase inhibitors, and Poly(ADP-ribose) polymerase (PARP) inhibitors are part of clinical trials or are preserved for specific sarcoma subtypes so far [13,14,15].

As a result, the alkylating agent ifosfamide is used in a large number of sarcoma patients. However, besides cytopenia, other serious treatment-related side effects, such as ifosfamide-induced neurotoxicity (IIN), limit treatment efficacy [16]. IIN occurs in about 10 to 40% of all patients treated with ifosfamide manifesting with somnolence, agitation, confusion, disorientation, hallucinations, or visual disturbances [17,18,19,20,21,22]. Symptoms usually appear no earlier than 12 h after the initiation of therapy and are usually self-limiting. However long-term consequences and fatal outcomes have been described [23,24,25,26]. Although the exact pathophysiological mechanisms for the development of IIN are not fully understood, substances with ifosfamide metabolism that can cross the blood–brain barrier, such as chloroacetaldehyde, have been discussed to be involved in this process [27,28,29]. Thiamine and methylene blue are used in clinical practice to treat and possibly prevent IIN, even if their benefits, especially for preventing IIN, have not been finally validated [30,31,32,33].

As ifosfamide remains an essential drug in sarcoma treatment, the reliable, pre-therapeutic identification of high-risk patients for developing IIN could lead to a reduction in treatment-related morbidity and improved outcomes. Various studies have identified factors that are associated with the development of IIN in different solid tumors, including cerebral metastases, impaired renal function, defined as a serum creatinine value greater than or equal to 1.2 mg/dL or hypalbuminemia [19,21,34]. For sarcoma patients, however, the available data are insufficient so far; in particular, for larger cohorts, investigations into the multitude of factors that can possibly influence the development of IIN are missing. Here, we retrospectively identified and further analyzed parameters associated with the occurrence of IIN in a larger cohort of sarcoma patients. Besides the identification of several risk factors for the development of IIN in sarcoma, this study shows, for the first time, that IIN serves as a prognostic and predictive factor in sarcoma.

## 2. Materials and Methods

### 2.1. Study Population

Patients with a histopathological confirmed diagnosis of sarcoma, treated at the Sarcoma Center of the University Hospital Tuebingen, were included in this observational study. Between January 2016 and October 2021, 172 patients (109 males and 63 females, with a mean age of 60.0 ± 14.9 years) with a diagnosis of sarcoma received systemic therapy with ifosfamide for at least one cycle (Appendix A). Treatment decisions were based exclusively on decisions made in an interdisciplinary tumor board, and ifosfamide was used in neoadjuvant, adjuvant, and palliative settings. The concept of the neoadjuvant treatment with ifosfamide was based on the reduction of the tumor burden and to enhance resectability [35]. In a palliative setting, ifosfamide was used especially for patients with highly symptomatic or rapid progressive disease [36]. The most commonly used protocol combines ifosfamide (administered at a dose of 3 g/m^2^ body surface area (BSA) continuously over 24 h for 3 days) with doxorubicin for three to six therapy cycles. Primary prophylaxis involved Mesna at a dose of 1000 mg/m^2^ body surface area (BSA) during treatment with ifosfamide (until 12 h after the completion of the last ifosfamide dose). In addition, 5 mg of olanzapine was administered once per day for 7 days. In the case of IIN grade 2 or higher, according to CTCAE, ifosfamide infusion was stopped and methylene blue (3 × 50 mg i.v.) combined with thiamine (200 mg i.v.) w administered. After the development of IIN, methylene blue and thiamine were given as a secondary prophylaxis in each subsequent cycle. The tumor characteristics were based on clinical staging, which was performed prior to the initiation of therapy with ifosfamide. Patient characteristics are shown in detail in Table 1. The study was approved by the IRB (ethics committee of the Faculty of Medicine of the Eberhard Karls University Tuebingen) of the University Hospital Tuebingen, and was conducted in accordance with the Declaration of Helsinki (reference number 163/2022BO2).

### 2.2. Collection of Data

For each patient included in the study, the following parameters were evaluated: sex, age, primary diagnosis, overall survival (OS) after initial histologic diagnosis of sarcoma, progression free survival (PFS) after exposure to ifosfamide, TNM classification, UICC stage, histological grading, concomitant cytotoxic agents, ifosfamide dose (mg/m^2^) per day, number of therapy cycles, infusion time, co-medication (at time of IIN), and secondary diagnoses. Additionally, body mass index (BMI), absolute neutrophile count (ANC), absolute lymphocyte count (ALC), hemoglobin (Hb) level, platelet (PLT) count, level of serum creatinine, sodium, calcium, alanine aminotransferase (GPT), aspartate aminotransferase (GOT), gamma-glutamyl transferase (GGT), and C-reactive protein were determined at the initiation of therapy with ifosfamide. The laboratory parameters were standardly determined before initiating treatment with ifosfamide and were frequently controlled over the course of therapy, especially at the occurrence of symptoms of an IIN. Ifosfamide-induced neurotoxicity was defined as an adverse effect directly caused by systemical treatment with ifosfamide, leading to alterations of the mental status ranging from retardation or confusion to hallucinations, seizures, or coma. IIN was determined and classified according to the Common Terminology Criteria for Adverse Events (CTCAE) version 5.0 guidelines. Patients were obligatorily assessed for neurological abnormalities based on a daily anamnesis and physical and neurological examination. Patients who showed corresponding symptoms in accordance with the CTCAE criteria were assigned into the subgroup “encephalopathy”. For these patients, the period until IIN manifested was determined. Patients who had no neuropsychiatric abnormalities during and in the short-term period after ifosfamide therapy were classified as “no encephalopathy”.

### 2.3. Data Analysis

Descriptive statistics were applied to characterize patients according to age, sex, TNM stage, histological grading, UICC stage, concomitant cytotoxic agents, ifosfamide dose per square meter per day, number of therapy cycles, infusion time, secondary diagnoses, and BMI. Patients with or without IIN were compared with respect to age, sex, BMI, primary diagnosis, TNM classification, histological grading, UICC stage, concomitant cytotoxic agents, number of therapy cycles, infusion time, co-medication and secondary diagnoses. The statistical significance of differences was analyzed using the chi-squared or Fisher’s exact test for categorical variables and the unpaired t-test for continuous variables. The predictive value of the predictive factors identified in this study were evaluated by examining the area under the receiver operator characteristic (ROC) with a confidence interval of 95%. Overall survival (OS) and progression free survival (PFS), including the median, were calculated using the Kaplan–Meier method. Continuous variables are presented as mean ± SD, and categorical variables are given by numbers and percentages. All statistical tests were considered statistically significant when *p* was below 0.05. Statistical analysis was performed using GraphPadPrism (v.9.1.2).

## 3. Results

### 3.1. Ifosfamide-Induced Neurotoxicity (IIN) Is Independent of Tumor Entity or Tumor Stages

In our study cohort, 172 sarcoma patients receiving systemic therapy with ifosfamide were analyzed (Appendix A). Reflecting the heterogeneous nature of sarcoma, several histopathological subtypes including liposarcoma (26.8%), undifferentiated pleomorphic sarcoma (18.6%), osteosarcoma (11.6%), and Ewing Sarcoma (10.5%) were included (Figure 1a). The majority of patients showed a limited disease stage without evidence of distant metastases (76.7%). During treatment with ifosfamide, 49 patients (28.5%) showed neuropsychiatric abnormalities, which were assessed as IIN. In 71.4% of patients, IIN was observed during the first cycle of treatment. The majority of patients developed symptoms of IIN in a time interval of 24 up to 48 h after the initiation of therapy. The mean time after that IIN manifested was 44 h (Appendix A). Moreover, 29 patients suffering from IIN received thiamine, 22 patients received methylene blue, and 4 patients were treated with haloperidol (Appendix A). While patients who received treatment with concomitant cytotoxic agents (e.g., doxorubicin, cisplatin, vincristine, or etoposide) did not show a higher incidence of IIN, the mean infusion time of ifosfamide was longer in patients with signs of IIN (15.65 vs. 11.04 h, *p* = 0.009, Table 2). In addition, a correlation was found between combined radio-chemotherapy, with ifosfamide acting as a radiosensitizing agent, and the frequency of IIN. In the group of sarcoma patients with IIN, a significantly lower proportion of patients underwent simultaneous local radiotherapy (10.2% vs. 35.8%, *p* < 0.0008).

When comparing additional treatment-related characteristics in sarcoma patients with and without IIN, gender or age showed no significant influence on the incidence of IIN (Table 2). Of note, patients receiving opioids and/or anticonvulsants showed a significantly higher rate of IIN. Interestingly, the occurrence of IIN was independent of UICC stages (*p* = 0.17), tumor grading (*p* = 0.17), tumor stage (*p* = 0.06), lymph node invasion (*p* = 0.22), and metastasis (*p* = 0.09) (Figure 1b–f). In our study cohort, treatment had to be modified in eight cases (4.7% of all patients treated with ifosfamide), in one of these ifosfamide had to be replaced with cyclophosphamide to due enormous adverse effects. No case of directly IIN-related death could be determined within this study.

### 3.2. Association of Routinely Determined Laboratory Markers and IIN

We analyzed routinely determined laboratory markers including the blood glucose level, CRP, creatinine, uric acid, natrium, and calcium, as well as GGT, GPT, GOT, LDH, and the occurrence of IIN in sarcoma patients. Patients presenting with IIN showed a higher CRP level (*p* = 0.004), higher natrium concentrations (*p* = 0.003), lower calcium level (*p* = 0.002), and higher GGT concentrations (*p* < 0.001) (Figure 2a–k). While IIN was independent of the absolute neutrophil count (ANC) and platelet (PLT) concentrations, patients exhibiting a lower absolute lymphocyte count (ALC), (*p* < 0.001) and lower hemoglobin level (Hb), (*p* < 0.001) more often showed signs of IIN (Figure 2l–o). The albumin level could retrospectively be determined in 31 out of 49 patients. Interestingly, with a mean of 3.45 g/dL (0.64 ± SD), patients with IIN showed no significant alterations regarding the serum albumin level.

### 3.3. Predictive Value of Clinical and Laboratory Findings and IIN in Sarcoma Patients

To investigate the predictive value of the clinical and laboratory findings associated with IIN in sarcomas, we performed a receiver operator characteristic (ROC) analysis. While the predictive value of radiotherapy, infusion time, CRP, calcium, natrium, and GGT (AUC 0.61–0.69, Figure 3a–f) showed only limited accuracy, with an AUC of 0.78 (95% CI: 0.7–0.86), decreased hemoglobin level, and lower ALC (AUC 0.71 (95% CI: 0.6–0.82), and showed the highest predictive value for IIN in sarcoma patients (Figure 3g,h).

Finally, we investigated the influence of IIN on the patient outcomes and overall survival (OS). Interestingly, patients presenting with IIN showed a significantly reduced progression free survival (PFS) compared with sarcoma patients without signs of IIN (median PFS 9 vs. 16 months, *p* < 0.001, Figure 3i). Moreover, the OS in sarcoma patients with IIN was significantly reduced compared with the patients without IIN (overall survival 18 months vs. 27 months; HR 2.04, 95% CI: 1.2-3.9, *p* < 0,001, Figure 3j). In addition, differences in OS were observed considering the degree of IIN defined via CTCAE (version 5.0). Remarkably, patients with severe neuropsychiatric abnormalities (grade 4) had significantly reduced overall survival (4 months) compared with patients with intermediate or milder expression of grade 3 (17 months) or grade 2 (22.5 months). Patients with CTCAE grade 1 showed the best OS probability (Figure 3k).

### 3.4. Prognostic Role of the Identified Risk Factors in Sarcoma Patients

As most of the identified risk factors for the development of IIN in our study cohort are associated with inflammation, we further investigated the prognostic role of of CRP, calcium, GGT, LDH, ALC, and Hb for PFS and OS in patients with and without IIN. While CRP and ALC were shown to be significantly associated with PFS in patients with IIN (Appendix A), in patients without signs of IIN, none of the identified factors were correlated with PFS (Appendix A). Interestingly, as shown in Figure 4a,f, the CRP and Hb levels predicted OS in patients with and without IIN, suggesting that these markers might be independently associated with a worse prognosis in sarcoma. The observation that inflammation and especially CRP and Hb level are associated with a worse outcome in sarcoma patients has been shown previously [37]. To further validate this observation, we investigated the prognostic role of ALC, Hb, LDH, and CRP in a cohort of 10 sarcoma patients without ifosfamide treatment (Appendix A, Appendix A). Of note, in these sarcoma patients, Hb and CRP were shown to be associated with a worse prognosis.

## 4. Discussion

In this observational study, we investigated the occurrence of IIN and predisposing factors in a cohort of sarcoma patients, in which ifosfamide still remains an indispensable anticancer drug in both first- and second-line treatment. IIN was observed in 28.5% of sarcoma patients receiving ifosfamide, which is comparable to previous studies in heterogeneous populations of cancer patients. Here, the occurrence of IIN has been described to be within a range of 15% to 30% [27,30,34,38,39].

Interestingly, in our study cohort, we did not observe an association between IIN and concomitant treatment. This is in contrast with previous studies showing that the concomitant or prior administration of cisplatin was associated with a higher risk of IIN [30,34]. Bearing in mind that the contemporary use of ifosfamide and cisplatin may result in increased cumulative nephrotoxicity, a higher risk for the development of IIN can be speculated [40]. However, in accordance with our results, data presented by Lo et al. failed to show a clear correlation between concomitant treatment with cisplatin and the development of IIN [41].

A further newly emerged aspect in our study is the influence of concurrent irradiation on the occurrence of IIN. To the best of our knowledge, a detailed analysis of the association between radiotherapy and the emergence of IIN is lacking so far. In our cohort of sarcoma patients, IIN was less frequently observed in the subgroup receiving irradiation in parallel. Of note, compared with non-irradiated patients, the mean age of patients undergoing combined radiochemotherapy was even higher (average age of patients with radiotherapy: 55.7 years versus average age of patients without radiotherapy: 49.1 years). Regarding the tumor entity, not all histological subtypes, for instance osteosarcoma, received combined radiochemotherapy. This might impact patient distribution according to this subgroup analysis. However, in accordance with our findings for the whole patient cohort, no clear differences regarding tumor size were found between these subgroups. Of note, the majority of patients treated with combined radiochemotherapy showed limited disease stages, as radiochemotherapy is mainly used in the case of locally advanced tumors lacking distant metastases. The observation that IIN was less frequently observed in sarcoma patients receiving irradiation in parallel might also be related to the respective treatment schedule of radiochemotherapy. According to the treatment regime used in these patients, ifosfamide was administered for two consecutive days for 4 h. Compared with the other treatment regimens used in our study, this treatment schedule used a reduced infusion time of ifosfamide. In contrast to our data showing a negative correlation with infusion time and risk for IIN, a shorter infusion time has previously described with a higher risk of IIN [30,42,43,44]. Of note, further prospective data in a homogenous patient cohort are needed to investigate this observation in more detail.

As renal function has recently been discussed as risk factor for the development of IIN, we investigated the serum creatinine level and uric acid concentration in our cohort of sarcoma patients. In contrast with previously published studies that considered an elevated serum creatinine level to be a risk factor for the occurrence of IIN [30,34,41,45], in our cohort, the serum creatinine as well as uric acid level were not associated with IIN. However, serum creatinine has been controversially discussed in this context. Although the serum levels differed significantly between patient groups with and without IIN, they often remained in the normal range [34]. In sarcoma patients, it has been discussed whether a tumor in the pelvic region may promote IIN due to alterations in renal perfusion [46]. In our patient population, there was no significant difference regarding tumor location in the pelvic region (patients with IIN: 32.7% vs. patients without IIN: 31.7%). Additionally, we evaluated routinely determined blood counts and laboratory parameters associated with IIN. We showed significant differences with regard to lymphocyte count, hemoglobin, sodium, calcium, GGT, and CRP levels. In contrast with the findings provided by Szabatura et al., we found lower hemoglobin levels in the patient cohort with IIN [34]. This is in line with data provided by Sweiss et al., in which anemia was assessed as a potential risk factor for the development of IIN [21]. In this context, a higher concentration of neurotoxic metabolites due to the lack of erythrocytes, which have an important transport function, has been discussed. However, corresponding studies for a more precise characterization are still pending.

Previously, electrolyte abnormalities have been defined as a minor risk factor for the development of IIN [45]. This includes changes in the serum sodium. We observed higher levels in the subpopulation of patients with IIN, although these remainws normal in most sarcoma patients.

In addition, we found significantly lower serum calcium levels in patients with IIN. Changes in the calcium itself have not been associated with the development of IIN so far. However, decreased calcium levels are associated with hypalbuminemia, which has been identified as a risk factor for IIN in numerous previous studies [21,34,38,41,45]. As ifosfamide is metabolized hepatically [30], changes in the liver enzymes have previously been observed in patients with and without IIN. In our cohort, we observed significantly higher levels of GGT in patients with IIN, which had not yet been described so far.

Interestingly, several factors identified in our study, including anemia, lymphopenia, thrombocytosis, and a high CRP level, reflect inflammatory markers. Although hypercalcemia is commonly described in advanced tumors and inflammation [47], we observed a lower calcium level to be associated with an inflammatory response and IIN. Of note, inflammatory cytokines, including IL-6, have been described to increase the parathyroid calcium-sensing receptor (CaSR), which reduces parathyroid hormone (PTH) secretion, resulting in inflammation-associated hypocalcemic hypoparathyroidism [48].

How inflammation-associated hypocalcemic hypoparathyroidism might influence IIN is unclear so far, and warrants further investigation. The observation that a low lymphocyte count was associated with an inflammatory phenotype and finally a reduced overall and progression free survival is in line with previous data, revealing lymphopenia as a relevant prognostic factor in several cancer types, including sarcoma [49,50,51,52].

As we observed that several inflammatory parameters act as prognostic markers in sarcoma patients with and without IIN, as well as without ifosfamide treatment, we conclude that inflammation represents an independent risk factor in sarcoma patients and is associated with a higher risk for the development of IIN. However, the underlying mechanism explaining the relationship between IIN and the inflammatory state is unclear so far. Several studies have shown that the permeability of the blood–brain barrier is increased during systemic inflammation. As a result, a rational link between inflammatory state and the risk of IIN might be speculated [53].

Interestingly, we demonstrated that the occurrence of IIN in sarcoma patients was associated with a survival disadvantage, even though the histopathological subgroup of sarcoma, as well as the UICC stages, tumor grading, and the occurrence of metastasis, were independent of the occurrence of IIN. In addition, patients with severe neuropsychiatric symptoms (CTCAE grade 4) appeared to have an even worse prognosis with regard to overall survival compared with those with mild symptoms. To what extent IIN acts as independent risk factor in sarcoma patients could not be determined in our study and warrants further investigation. Sweiss et al. discussed that the early discontinuation of therapy with ifosfamide due to IIN may compromise survival outcome [21]. However, we have no evidence that the majority of patients required an interruption of therapy because of IIN. Possible factors influencing this may be a poor general condition or more comorbidities in the patient group with an IIN.

In summary, our work provides numerous new insights into which factors may contribute to the development of IIN, specifically in sarcoma patients. By identifying patients with a high-risk profile for IIN, our study might help to identify and reduce therapy-associated morbidity. This appears to be of particular importance in sarcoma patients, as there are limited treatment options available so far, besides ifosfamide. Nevertheless, our study has some limitations, such as the retrospective study design and temporal incongruence in the collection of blood values in patients with and without IIN. The observation that an inflammatory state is associated with an increased risk of IIN can be used prospectively to further investigate the relationship between inflammation and IIN. Moreover, the easily measured blood markers found to predict IIN in our study can be used for clinical decision making.

## Figures and Tables

**Figure 1 jcm-11-05798-f001:**
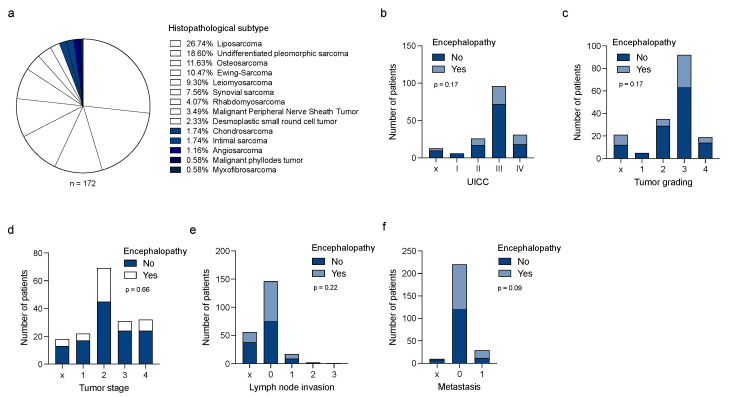
Clinical characteristics in STS patients with IIN. (**a**) Histopathological subtypes in the sarcoma study cohort. (**b**) Number of sarcoma patients with or without IIN with regard to UICC classification, tumor grading (**c**), tumor stage (**d**), lymph node invasion, (**e**) and the occurrence of metastasis (**f**). UICC = Union for International Cancer Control; IIN = ifosfamide-induced neurotoxicity.

**Figure 2 jcm-11-05798-f002:**
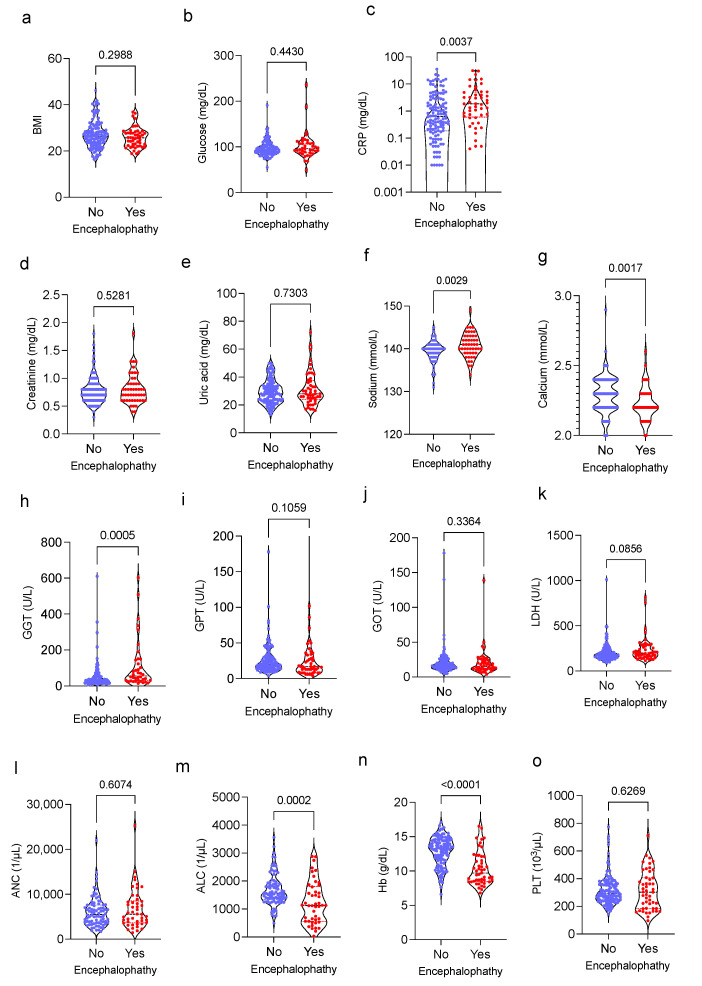
Correlation of clinical and laboratory parameters and IIN. (**a**) Correlation of BMI and IIN. (**b**) Association of glucose level, CRP (**c**), creatinine level (**d**), uric acid (**e**), sodium (**f**), calcium (**g**), GGT (**h**), GPT (**i**), GOT (**j**) LDH (**k**), ANC (**l**), ALC (**m**), Hb (**n**), and PLT count (**o**) and the occurrence of IIN. BMI = body mass index; IIN = ifosfamide-induced neurotoxicity; CRP = C-reactive protein; GGT = gamma-glutamyl transferase, GPT = glutamic pyruvic transaminase = alanine aminotransferase; GOT = glutamic oxaloacetic transaminase = aspartate aminotransferase; LDH = lactate dehydrogenase; ANC = absolute neutrophil count; ALC = absolute lymphocyte count; Hb = hemoglobin level; PLT = platelet.

**Figure 3 jcm-11-05798-f003:**
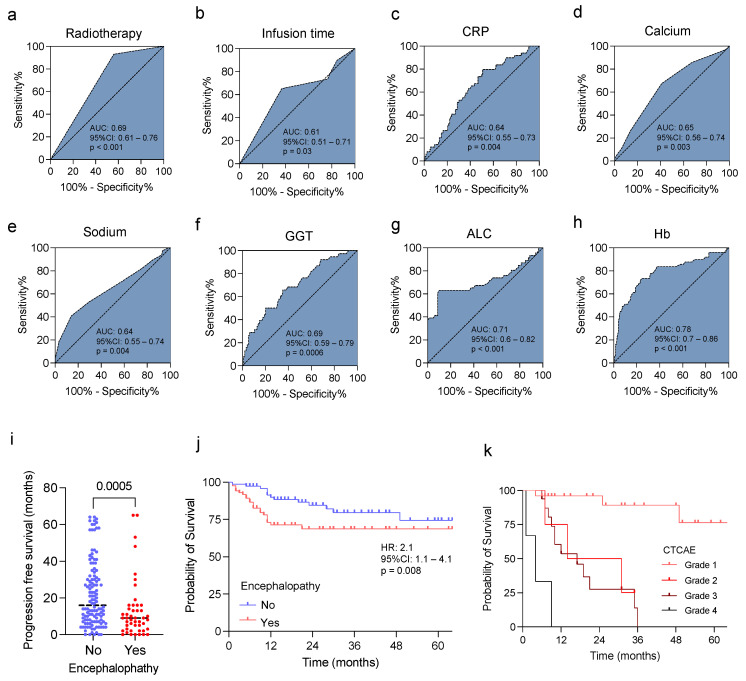
Predictive value of risk factors and influence of IIN on patient outcomes in STS. Predictive values of radiotherapy (**a**), ifosfamide infusion time (**b**), CRP (**c**), calcium (**d**), sodium (**e**), GGT (**f**), ALC (**g**), and Hb level (**h**) for the development of IIN were analyzed using ROC analysis. (**i**) Progression free survival in sarcoma patients with and without IIN. (**j**) Kaplan–Meier curve estimates of survival in sarcoma patients with and without IIN. (**k**) Kaplan–Meier curves showing the influence of CTCAE grade and probability of survival. IIN = ifosfamide-induced neurotoxicity; CRP = C-reactive protein; GGT = gamma-glutamyl transferase; ALC = Absolute lymphocyte count; Hb = hemoglobin level; ROC = receiver operator characteristic; CTCAE = Common Terminology Criteria for Adverse Events.

**Figure 4 jcm-11-05798-f004:**
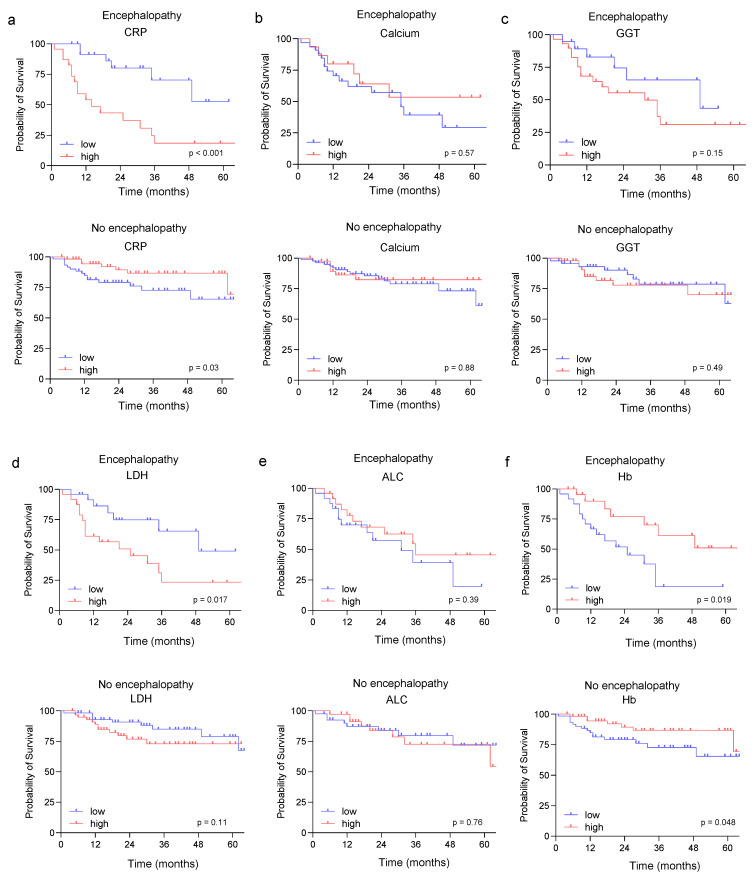
Prognostic role of the identified risk factors in sarcoma patients. Kaplan–Meier curves showing the influence of CRP (**a**), calcium (**b**), GGT (**c**), LDH (**d**), ALC (**e**), and HB levels (**f**) in patients with and without IIN and probability of survival. IIN = ifosfamide-induced neurotoxicity; CRP = C-reactive protein; GGT = gamma-glutamyl transferase; ALC = absolute lymphocyte count; Hb = hemoglobin level; LDH = lactate dehydrogenase.

**Table 1 jcm-11-05798-t001:** Patient characteristics.

Patient Characteristics	Total	(%)
	(n = 172)	
**Gender**		
female, sex (n)	63	36.6
**Age**	18 to 76	
Age in years, mean–yr. ± SD	60.0 +/− 14.9	
**TNM classification**		
Stage		
Tx	18	10.4
T1	22	12.8
T2	69	40.1
T3	31	18.1
T4	32	18.6
Node		
Nx	56	32.6
N0	96	55.8
N1	17	9.8
N2	2	1.2
N3	1	0.6
Metastasis		
Mx	10	5.8
M0	132	76.7
M1	30	17.5
**Histological grading, n (%)**		
Gx	21	12.2
G1	5	2.9
G2	35	20.3
G3	92	53.5
G4	19	11.1
**Histology**		
Angiosarcoma	2	1.2
Chondrosarcoma	3	1.7
Desmoplastic small round cell tumor	4	2.3
Ewing Sarcoma	18	10.5
Intimal sarcoma	3	1.7
Leiomyosarcoma	16	9.3
Liposarcoma	46	26.7
Malignant Peripheral Nerve Sheath Tumor	6	3.5
Malignant phylloides tumor	1	0.6
Myxofibrosarcoma	1	0.6
Osteosarcoma	20	11.6
Rhabdomyosarcoma	7	4.1
Synovial sarcoma	13	7.6
Undifferentiated pleomorphic sarcoma	32	18.6
**UICC stage**		
X	13	7.6
I	6	3.5
II	26	15.1
III	96	55.8
IV	31	18.0
**Therapy line at time-point of treatment with ifosfamide**		
1st line	166	96.5
2nd line	5	2.9
>2nd line	1	0.6
**Treatment setting**		
Neoadjuvant	107	62.2
Adjuvant	43	25.0
Neoadjuvant + adjuvant	16	9.3
Palliative (primarily)	25	14.5
Combined radiochemotherapy	49	28.5
Surgery	154	89.5
**Concurrent therapy**		
Monotherapy	52	30.2
Anthracycline-based	77	44.8
Platinum- and Anthracycline-based	15	8.7
1 combinatorial treatment partner	78	45.3
2 combinatorial treatment partners	19	11.0
>2 combinatorial treatment partners	23	13.4
**Ifosfamide daily dose (mg/m²/day)**		
3000	172	100.0
**Number of therapy cycles**		
1–3	95	55.2
4–6	67	39.0
>6	10	5.8
**Infusion time**		
1 h	23	13.4
2 h	15	8.7
3 h	4	2.3
4 h	53	30.8
24 h	77	44.8
**Comorbidities**		
hypertension	52	30.2
hypothyroidism	15	8.7
diabetes mellitus	13	7.6
chronic kidney disease	8	4.7
bronchial asthma	7	4.1

**Table 2 jcm-11-05798-t002:** Treatment characteristics.

Encephalopathy	No (n = 123)	Yes (n = 49)	*p*-Value
**Treatment**			
concurrent doxorubicin	47 (38.2%)	30 (61.2%)	0.06
concurrent cisplatin	10 (8.1%)	5 (10.2%)	0.67
concurrent vincristine	19 (15.4%)	7 (14.3%)	0.85
concurrent etoposide	18 (14.6%)	6 (12.2%)	0.68
**Infusion time**			
1 h	18 (14.6%)	5 (10.2%)	0.007
2 h	9 (7.3%)	6 (12.2%)	
3 h	2 (1.6%)	2 (4.1%)	
4 h	47 (38.2%)	6 (12.2%)	
24 h	47 (38.2%)	30 (61.2%)	
Mean	11.04	15.65	0.009
SD	10.28	10.62	
**Cycle**			
1st	x	35 (71.4%)	
2nd and 3rd	x	6 (12.3%)	
>3rd	x	8 (16.3%)	
**Radiotherapy**			
Yes	44 (35.8%)	5 (10.2%)	0.0008
No	79 (64.2%)	44 (89.8%)	
**Co-medication**			
opioids	21 (17.1%)	20 (40.8%)	0.85
anticonvulsant	11 (8.9%)	13 (26.5%)	
antidepressant	9 (7.3%)	7 (14.3%)	
anxiolytic	1 (0.8%)	1 (2.0%)	
neuroleptic	1 (0.8%)	0 (0%)	
**Age**	49.9 +/− 14.09	53.61 +/− 16.51	0.14
**Sex**			
Male	81 (65.9%)	28 (57.1%)	0.284
Female	42 (34.1%)	21 (42.9%)	

## Data Availability

The data presented in this study are available on request from the corresponding author.

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
