# Peer review of "Inflammatory Surrogate Parameters for Predicting Ifosfamide-Induced Neurotoxicity in Sarcoma Patients"

_jcm, 2022, doi:10.3390/jcm11195798_

Round 1

Reviewer 1 Report

The authors of this retrospective study of ifosfamide-induced neurotoxicity / encephalopathy (IIN) have identified an association between a number of easily measured blood markers, CRP, lymphopenia and anaemia, in particular, and the risk of IIN.  There was also a correlation with prolonged ifosfamide infusion times, i.e. 24 hours compared with 4 hours, and the use of opioids and anticonvulsants.  What is still lacking is an explanation for these associations.  However, the relationship between the increased risk of IIN and both progression-free and overall survival provides a clue.  The mistake is in claiming to have found a new prognostic marker for sarcomas.  The fact is that anaemia, lymphopenia, neutrophilia, thrombocytosis and high CRP are all markers of the inflammatory response seen in cancer and infection.  There is also a relationship between the inflammatory response and calcium but it is complex. The inflammatory state is associated with immunosuppression and numerous publications have demonstrated a relationship between inflammatory response and poorer prognosis in multiple cancers.  The relationship with lymphocyte count is particularly instructive, in that lymphopenia is an adverse prognostic marker in cancer, including sarcoma.  Various studies have looked at absolute lymphocyte count or, more commonly, the neutrophil/lymphocyte ratio.  A significant proportion of patients with IIN in this study had an absolute lymphocyte count of <1 x 109/l.  Similarly a significant proportion had a haemoglobin of <10 g/dl. Regarding lymphopenia and prognosis in sarcomas the following publications are pertinent:

[Ray-Coquard I, et al. Lymphopenia as a prognostic factor for overall survival in advanced carcinomas, sarcomas, and lymphomas. Cancer Res. 2009;69(13):5383-5391

Szkandera J, Absenger G, Liegl-Atzwanger B, et al. Elevated preoperative neutrophil/lymphocyte ratio is associated with poor prognosis in soft-tissue sarcoma patients. Br J Cancer. 2013;108(8):1677-1683.

Brewster R, et al. Evaluation of Absolute Lymphocyte Count at Diagnosis and Mortality Among Patients With Localized Bone or Soft Tissue Sarcoma. JAMA Netw Open. 2021;4(3):e210845.

Jiang M, et al. Prognostic Value of Pretreated Blood Inflammatory Markers in Patients with Bone Sarcoma: A Meta-Analysis. Dis Markers. 2021;2021:8839512.]

So, what is lacking is an explanation for the apparent relationship between inflammatory state and IIN.  Perhaps this can be explained by the complex changes that occur in the blood-brain barrier in response to inflammation. 

[Galea I. The blood-brain barrier in systemic infection and inflammation. Cell Mol Immunol. 2021;18(11):2489-2501.

Danielski LG et al.  Brain barrier breakdown as a cause and consequence of neuroinflammation in sepsisMol Neurobiol. 2018;55(2):1045–1053]

The manuscript could be improved by acknowledging the relationship between inflammatory markers and prognosis in sarcomas and reviewing the possible relationship between inflammation and ifosfamide-induced neurotoxicity, possibly related to alterations in the blood-brain barrier.  The claim that IIN is a novel predictive and prognostic marker in sarcomas should be deleted.  The fact that elevated inflammatory markers are associated with an increased risk of IIN is nevertheless valuable information and can certainly be used prospectively, both to study this relationship in more detail but also to aid decision making.

There is a major error in Fig 3j, the legend is incorrect – the blue curve is No encephalopathy, the red curve Yes – colours as elsewhere in Fig 2 and Fig 3i. 

In Table 1 it should read UICC stage, not stadium

In Methods 2.2:  It is more usual to say “lymphocyte count” rather than lymphocytes, also applies to neutrophil, platelet, etc.  The same applies to the sarcoma subtypes in Results 3.1 – liposarcoma, not liposarcomas etc.

The authors of this retrospective study of ifosfamide-inducted neurotoxicity / encephalopathy (IIN) have identified an association between a number of easily measured blood markers, CRP, lymphopenia and anaemia, in particular, and the risk of IIN.  There was also a correlation with prolonged ifosfamide infusion times, i.e. 24 hours, and the use of opioids and anticonvulsants.  What is still lacking is an explanation for these associations.  However, the relationship between the increased risk of IIN and both progression-free and overall survival provides a clue.  The mistake is in claiming to have found a new prognostic marker for sarcomas.  The fact is that anaemia, lymphopenia, neutrophilia, thrombocytosis and high CRP are all markers of the inflammatory response seen in cancer and infection.  There is also a relationship between the inflammatory response and calcium but it is complex. The inflammatory state is associated with immunosuppression and multiple publications have demonstrated a relationship between inflammatory response and poorer prognosis in multiple cancers.  The relationship with lymphocyte count is particularly instructive, in that lymphopenia is an adverse prognostic marker in cancer, including sarcoma.  Various studies have looked at absolute lymphocyte count or, more commonly, the neutrophil/lymphocyte ratio.  A significant proportion of patients with IIN in this study had an absolute lymphocyte count of <1 x 109/l.  Similarly a significant proportion had a haemoglobin of <10 g/dl. Regarding lymphopenia and prognosis in sarcomas the following publications are pertinent:

[Ray-Coquard I, et al. Lymphopenia as a prognostic factor for overall survival in advanced carcinomas, sarcomas, and lymphomas. Cancer Res. 2009;69(13):5383-5391

Szkandera J, Absenger G, Liegl-Atzwanger B, et al. Elevated preoperative neutrophil/lymphocyte ratio is associated with poor prognosis in soft-tissue sarcoma patients. Br J Cancer. 2013;108(8):1677-1683.

Brewster R, et al. Evaluation of Absolute Lymphocyte Count at Diagnosis and Mortality Among Patients With Localized Bone or Soft Tissue Sarcoma. JAMA Netw Open. 2021;4(3):e210845.

Jiang M, et al. Prognostic Value of Pretreated Blood Inflammatory Markers in Patients with Bone Sarcoma: A Meta-Analysis. Dis Markers. 2021;2021:8839512.]

So, what is lacking is an explanation for the apparent relationship between inflammatory state and IIN.  Perhaps this can be explained by the complex changes that occur in the blood-brain barrier in response to inflammation. 

[Galea I. The blood-brain barrier in systemic infection and inflammation. Cell Mol Immunol. 2021;18(11):2489-2501.

Danielski LG et al.  Brain barrier breakdown as a cause and consequence of neuroinflammation in sepsisMol Neurobiol. 2018;55(2):1045–1053]

The manuscript could be improved by acknowledging the relationship between inflammatory markers and prognosis in sarcomas and reviewing the possible relationship between inflammation and ifosfamide-induced neurotoxicity, possibly related to alterations in the blood-brain barrier.  The claim that IIN is a novel predictive and prognostic marker in sarcomas should be deleted.  The fact that elevated inflammatory markers are associated with an increased risk of IIN is nevertheless valuable information and can certainly be used prospectively, both to study this relationship in more detail but also to aid decision making.

There is a major error in Fig 3j, the legend is incorrect – the blue curve is No encephalopathy, the red curve Yes – colours as elsewhere in Fig 2 and Fig 3i. 

In Table 1 it should read UICC stage, not stadium

In Methods 2.2:  It is more usual to say “lymphocyte count” rather than lymphocytes, also applies to neutrophil, platelet, etc.  The same applies to the sarcoma subtypes in Results 3.1 – liposarcoma, not liposarcomas etc.

Author Response

The authors of this retrospective study of ifosfamide-induced neurotoxicity / encephalopathy (IIN) have identified an association between a number of easily measured blood markers, CRP, lymphopenia and anaemia, in particular, and the risk of IIN.  There was also a correlation with prolonged ifosfamide infusion times, i.e. 24 hours compared with 4 hours, and the use of opioids and anticonvulsants.  What is still lacking is an explanation for these associations.  However, the relationship between the increased risk of IIN and both progression-free and overall survival provides a clue.  The mistake is in claiming to have found a new prognostic marker for sarcomas.  The fact is that anaemia, lymphopenia, neutrophilia, thrombocytosis and high CRP are all markers of the inflammatory response seen in cancer and infection.  There is also a relationship between the inflammatory response and calcium but it is complex. The inflammatory state is associated with immunosuppression and numerous publications have demonstrated a relationship between inflammatory response and poorer prognosis in multiple cancers.  The relationship with lymphocyte count is particularly instructive, in that lymphopenia is an adverse prognostic marker in cancer, including sarcoma.  Various studies have looked at absolute lymphocyte count or, more commonly, the neutrophil/lymphocyte ratio.  A significant proportion of patients with IIN in this study had an absolute lymphocyte count of <1 x 109/l.  Similarly a significant proportion had a haemoglobin of <10 g/dl. Regarding lymphopenia and prognosis in sarcomas the following publications are pertinent:

[Ray-Coquard I, et al. Lymphopenia as a prognostic factor for overall survival in advanced carcinomas, sarcomas, and lymphomas. Cancer Res. 2009;69(13):5383-5391

Szkandera J, Absenger G, Liegl-Atzwanger B, et al. Elevated preoperative neutrophil/lymphocyte ratio is associated with poor prognosis in soft-tissue sarcoma patients. Br J Cancer. 2013;108(8):1677-1683. 

Brewster R, et al. Evaluation of Absolute Lymphocyte Count at Diagnosis and Mortality Among Patients With Localized Bone or Soft Tissue Sarcoma. JAMA Netw Open. 2021;4(3):e210845. 

Jiang M, et al. Prognostic Value of Pretreated Blood Inflammatory Markers in Patients with Bone Sarcoma: A Meta-Analysis. Dis Markers. 2021;2021:8839512.] 

So, what is lacking is an explanation for the apparent relationship between inflammatory state and IIN.  Perhaps this can be explained by the complex changes that occur in the blood-brain barrier in response to inflammation.  

[Galea I. The blood-brain barrier in systemic infection and inflammation. Cell Mol Immunol. 2021;18(11):2489-2501. 

Danielski LG et al.  Brain barrier breakdown as a cause and consequence of neuroinflammation in sepsis. Mol Neurobiol. 2018;55(2):1045–1053]

The manuscript could be improved by acknowledging the relationship between inflammatory markers and prognosis in sarcomas and reviewing the possible relationship between inflammation and ifosfamide-induced neurotoxicity, possibly related to alterations in the blood-brain barrier.  The claim that IIN is a novel predictive and prognostic marker in sarcomas should be deleted.  The fact that elevated inflammatory markers are associated with an increased risk of IIN is nevertheless valuable information and can certainly be used prospectively, both to study this relationship in more detail but also to aid decision making. 

We thank the reviewer for raising the helpful, important and valuable comments. Indeed several factors identified in our study including anaemia, lymphopenia, thrombocytosis and high CRP level are markers of an inflammatory response.

Although hypercalcemia is commonly described in advanced tumors and inflammation (Zagzag et al., CA: A Cancer Journal for Clinicians, 2018) we observed lower calcium level to be associated with an inflammatory response and IIN. Of note, inflammatory cytokines including IL-6 have been described to increase parathyroid calcium-sensing receptor (CaSR), which reduces parathyroid hormone (PTH) secretion resulting in an inflammation-associated hypocalcemic hypoparathyroidism (Klein at al., Biomolecules 2018). How an inflammation-associated hypocalcemic hypoparathyroidism might influences IIN is unclear so far and warrants further investigation.

We agree with the reviewer that the observation that a low lymphocyte count was associated with an inflammatory phenotype and finally a reduced overall and progression free survival is in line with previous data revealing lymphopenia as relevant prognostic factors in several cancer types including sarcoma (Ray-Coquard I, et al., Cancer Res. 2009; Szkandera J et al., Br J Cancer. 2013; Brewster R et al., JAMA Netw Open. 2021; Jiang M et al., Dis Markers. 2021).

However, the underlying mechanism explaining the relationship of IIN and the inflammatory state is unclear so far. We thank the expert for the insightful note that several studies indeed have shown that permeability of the blood brain barrier is increased during systemic inflammation. As a result a rational link between inflammatory state and the risk of IIN might be speculated (Galea I et al., Cell Mol Immunol. 2021; Danielski LG et al, Mol Neurobiol. 2018).

To what extent IIN acts as independent risk factor in sarcoma patients cannot be determined in our study and warrants further investigation. We apologize for this overstatement and amended the manuscript accordingly.

Finally, we completely agree with the reviewer that our observation that an inflammatory state is associated with an increased risk of IIN is a valuable information and can be used prospectively to further investigate relationship of inflammation and IIN. Moreover, the easily measured blood markers found to predict IIN in our study can be used for clinical decision making.

As a result the revised manuscript was amended as follows.

There is a major error in Fig 3j, the legend is incorrect – the blue curve is No encephalopathy, the red curve Yes – colours as elsewhere in Fig 2 and Fig 3i.  

We thank the reviewer for bringing up this important point. The manuscript was changed accordingly.

In Table 1 it should read UICC stage, not stadium

We agree with the important notion of the reviewer. According to the suggestions of the expert we amended Table 1.

In Methods 2.2:  It is more usual to say “lymphocyte count” rather than lymphocytes, also applies to neutrophil, platelet, etc.  The same applies to the sarcoma subtypes in Results 3.1 – liposarcoma, not liposarcomas etc.

We thank the reviewer for raising this important aspect and apologize for not stating clearly enough. Accordingly, the manuscript was amended.

Reviewer 2 Report

This manuscript aimed to assess risk factors associated with ifosfamide-induced neurotoxicity in patients with STS. The authors performed a retrospective cohort study including all patients who received ifosfamide-based chemotherapy in a single center. The manuscript was well written; however, some questions arose after reading:

1.    I found it remarkable that a substantial part of the patients who received chemotherapy, had a primary tumor. In a lot of centers, chemotherapy is not standard of care in primary STS, but is mainly offered in a palliative/metastatic setting. Could the authors please elaborate a bit more about the indication for chemotherapy and protocol in their center?

2.    It would also be insightful to add a flowchart including the number of STS patients treated in your center during the inclusion period and the percentage of patients receiving chemotherapy.

3.     The title and introduction suggest that only soft tissue sarcomas were included in this study, however in table 1 also chondrosarcoma, Ewing sarcoma and osteosarcoma were included in the study. I would suggest to exclude those tumors.

4.    How is IIN defined? Is this an outcome which could be well assessed retrospectively based on patient reports? Furthermore, how is co-medication defined? Is it also possible that the medication was prescribed after the development of IIN? At what time point is co-medication defined?

5.    Why did you not use time to event analysis for IIN? Could censoring have influenced your results? Shouldn’t INN be included as time varying variable?

6.    Could you stratify the baseline characteristics based on encephalopathy (yes/no)? It would be of interest to see what the association is between age, grade, size, stage etc. and INN.

7.    At what time point are the laboratory parameters performed? Could you state this in the methods section. It could not be a predictive factor if the laboratory parameters were performed after onset of IIN.

8.    How would you explain that INN was less frequently observed in the subgroup of patients who received irradiation in parallel? Are the patients who received RTX and patients who did not receive RTX comparable in terms of stage, performance, age, tumor type/size?

9.    This study uses words that suggest causal inference such as ‘comitant use of opioids or anticonvulsants affected the risk of developing INN’. However, this study aimed to assess risk factors/prognostic factors. For causal inference another study design would be more appropriate (e.g. case-control study matching patients with and without encephalopathy and see how many received opioids/or other parameter of interest). Or at least multivariable analysis adjusting for confounders should be performed. However, since this study aimed to assess predictors, wording that suggest causal relationship should be avoided, especially since the time points are exposure and outcome are poorly defined and the parameters and outcomes are not ‘hard’ parameters.

10. Could you also describe what the median follow-up was of the patients (stratified by INN), what the median time to development of INN was, how INN was treated/handled, whether mortality was INN related?

Author Response

This manuscript aimed to assess risk factors associated with ifosfamide-induced neurotoxicity in patients with STS. The authors performed a retrospective cohort study including all patients who received ifosfamide-based chemotherapy in a single center. The manuscript was well written; however, some questions arose after reading:

I found it remarkable that a substantial part of the patients who received chemotherapy, had a primary tumor. In a lot of centers, chemotherapy is not standard of care in primary STS, but is mainly offered in a palliative/metastatic setting. Could the authors please elaborate a bit more about the indication for chemotherapy and protocol in their center?

We thank the reviewer for this important comment. In this patient cohort, approximately 75% of all patients showed limited disease stages with the lack of lymph node or distant metastases. Most of these patients received a chemotherapy with ifosfamide in a neoadjuvant setting. Treatment decisions were based on a consent made in an interdisciplinary tumor board representing one central element of the sarcoma center of the University Hospital Tuebingen.

The concept of the neoadjuvant treatment in this patients is to reduce tumor burden and enhance respectability. This is especially important for high-grade tumors comprising more than half of all patients in our study population. Furthermore, ifosfamide does not only represent a component of systemical tumor treatment, but also acts as a radiosensitizer.

The use of ifosfamide in a neoadjuvant setting is supported by results of a study performed by Gronchi and colleagues demonstrating a survival benefit of ifosfamide in combination with epirubicin in comparison to other chemotherapeutic treatment regimens (Gronchi et al., Lancet Onocoloy 2017). In an adjuvant setting, chemotherapy with ifosfamide is commonly used to reduce the risk for local and/or distant metastasis. However, available data from clinical trials investigating the benefit of an adjuvant use of ifosfamide are conflicting, especially with regard to overall survival (Sarcoma Meta-analysis Collaboration, Lancet 1997; Woll et al., Journal of Clinical Oncology 2007; Pervaiz et al., Cancer 2008; O’Connor et al., Journal of Clinical Oncology 2008). In a palliative setting, ifosfamide can be used especially for patients with highly symptomatic or rapid progressive disease (van Oosterom et al, European Journal of Cancer 2002; Sleijfer et al., European Journal of Cancer 2010).

The most commonly used protocol in our study combines ifosfamide (administered at a dose of 3 g/m2 continuouslyover 24 hours for 3 days) with doxorubicin for three up to six therapy cycles.

To reflect this background information, the manuscript was amended accordingly.

It would also be insightful to add a flowchart including the number of STS patients treated in your center during the inclusion period and the percentage of patients receiving chemotherapy.

We thank the reviewer for raising this important issue. A flow chart contrasting sarcoma patients that were treated with and without chemotherapy during the inclusion period is depicted in supplementary figure 1.

The title and introduction suggest that only soft tissue sarcomas were included in this study, however in table 1 also chondrosarcoma, Ewing sarcoma and osteosarcoma were included in the study. I would suggest to exclude those tumors.

We thank the reviewer for this insightful advice and fully agree with his or her statement. The tumor entities osteosarcoma, chondrosarcoma and Ewing sarcoma that had comprised 23.8% of all tumors analyzed. Since an exclusion of these tumor entities reduce the statistical power of our analysis we decided to finally include these patients in this study. Of note, we amended the title of study and the corresponding passages of the manuscript accordingly. 

How is IIN defined? Is this an outcome which could be well assessed retrospectively based on patient reports? Furthermore, how is co-medication defined? Is it also possible that the medication was prescribed after the development of IIN? At what time point is co-medication defined?

We thank the reviewer for raising this issue. Ifosfamide-induced neurotoxicity was defined as an adverse effect that is directly caused by systemical treatment with ifosfamide leading to diverse alterations of the mental status of a patient with symptoms ranging from retardation, confusion to hallucinations, seizures and even coma. Since patients treated with ifosfamide are at risk of developing these symptoms, patients are obligatorily assessed for neurological abnormalities based on a daily anamnesis and physical and neurological examination. The respective findings are seamlessly integrated into an electronic medical documentation.

Co-medication in this study is defined as medication which is administered simultaneously to the application of ifosfamide. Most of this co-medication represents drugs to reduce nociceptive or neuropathic pain originated from an invasive tumor disease.  By means of the electronic medical documentation implemented in our tumor center, co-medication which is applied to the patients is carefully documented and self-medication during chemotherapy with ifosfamide is thought to be avoided. Thus, an existing co-medication could easily be evaluated at the exact day of IIN. To reflect these results, the manuscript was amended accordingly.

Why did you not use time to event analysis for IIN? Could censoring have influenced your results? Shouldn’t INN be included as time varying variable?

We thank the reviewer for pointing us towards this important issue. As described below, we additionally analyzed our patient cohort regarding the chronological occurrence of symptoms that are consistent with IIN.

Occurrence of IIN

Mean time until occurrence of IIN (hours)

43.97

Median time until occurrence of IIN (hours)

40.00

n

%

< 24 hours until occurrence of IIN

9

24.32

24 - 48 hours until occurrence of IIN

15

40.54

49 - 72 hours until occurrence of IIN

9

24.32

> 72 hours until occurrence of IIN

4

10.81

Of note, the majority of patients developed symptoms of IIN in a time interval of 24 up to 48 hours after the initiation of therapy. The mean time after that IIN became manifest was 44 hours which is in line with previously reports.

Treatment with ifosfamide in our institution is carefully supervised and only performed for inpatients. Additionally, patients are closely monitored up to one day after completing one therapy cycle.

To reflect these results, besides including our new data in Supplementary Table 1, the manuscript was amended accordingly.

Could you stratify the baseline characteristics based on encephalopathy (yes/no)? It would be of interest to see what the association is between age, grade, size, stage etc. and INN.

We thank the reviewer for raising this important issue. Since the majority of the patients included into this retrospective study were treated with ifosfamide as first-line therapy and most of the patients developed symptoms of an IIN within the first therapy cycle, characteristics such as age, tumor grade, size and stage were surveyed and are depicted as baseline characteristics in Figure 1. Of note, as described in Fig. 1 and Table 1-2, no relationship between the occurrence of IIN and age, gender, tumor size or stage could be found.

At what time point are the laboratory parameters performed? Could you state this in the methods section. It could not be a predictive factor if the laboratory parameters were performed after onset of IIN.

We thank the reviewer for pointing us towards this important issue. Laboratory parameters were frequently assessed over the course of therapy, among others due to potential nephrotoxicity. Based on this, it was possible to use current blood values for our investigation. To reflect this, the manuscript was amended accordingly.

How would you explain that INN was less frequently observed in the subgroup of patients who received irradiation in parallel? Are the patients who received RTX and patients who did not receive RTX comparable in terms of stage, performance, age, tumor type/size?

We thank the reviewer for raising this important issue. The finding that occurrence of IIN was significantly lower in the subgroup of patients receiving simultaneous irradiation was unexpected. One might speculate that this observation is related to differences in infusion time and days on treatment. Interestingly, the subgroup of patients receiving radiochemotherapy was older (average age of patients with radiotherapy: 55.7 years versus average age of patients without radiotherapy: 49.1 years); although no correlation of age and occurrence of IIN was found in the entire patient cohort. Since patients with some subtypes of sarcoma (e.g. osteosarcoma) did not receive combined radiochemotherapy, patients suffering from liposarcoma in our study population have received ifosfamide more frequently as a radiosensitizer. However, we didn’t observe a correlation of tumor histology and the occurrence of IIN. In accordance with our findings for the whole patient cohort, no differences regarding tumor size were observed in between these subgroups.

This study uses words that suggest causal inference such as ‘comitant use of opioids or anticonvulsants affected the risk of developing INN’. However, this study aimed to assess risk factors/prognostic factors. For causal inference another study design would be more appropriate (e.g. case-control study matching patients with and without encephalopathy and see how many received opioids/or other parameter of interest). Or at least multivariable analysis adjusting for confounders should be performed. However, since this study aimed to assess predictors, wording that suggest causal relationship should be avoided, especially since the time points are exposure and outcome are poorly defined and the parameters and outcomes are not ‘hard’ parameters.

We thank the reviewer for pointing us towards this important issue. The text passages are modified and highlighted, respectively.

Could you also describe what the median follow-up was of the patients (stratified by INN), what the median time to development of INN was, how INN was treated/handled, whether mortality was INN related?

We thank the reviewer for this rightful and important comment.

We again analyzed our patient cohort regarding the above-mentioned characteristics.

Median follow-up (months)

IIN

without IIN

Time to progression

8

17

Time to death

16

27

Mean follow-up (months)

Time to progression

11.00

22.05

Time to death

22.28

28.41

Duration of IIN

Mean duration of IIN (days)

3.17

Median duration of IIN (days)

3.00

Occurrence of IIN

Mean time until occurrence of IIN (hours)

43.97

Median time until occurrence of IIN (hours)

40.00

n

%

< 24 hours until occurrence of IIN

9

24.32

24 - 48 hours until occurrence of IIN

15

40.54

49 - 72 hours until occurrence of IIN

9

24.32

> 72 hours until occurrence of IIN

4

10.81

Treatment of IIN

n

%

Administration of thiamine

29

78.38

Administration of methylene blue

22

59.46

Administration of haloperidol

4

10.81

Of note, the median follow-up for PFS and OS was significantly longer in the patient cohort without symptoms of an IIN. The median duration of IIN was 3 days in this study and the majority of the patients developed symptoms of an IIN approximately 40 hours after the first administration of ifosfamide.

80% of patients developing IIN received any kind of treatment, with thiamine being the most frequent therapeutic approach, especially in cases of mild or moderate symptoms. This supportive medication was given intravenously three times a day until the symptoms dissolved.

No case of directly IIN-related death could be determined within this study, however treatment had to be modified in 8 cases, in one of these cases therapy with ifosfamide had to be replaced with cyclophosphamide to due enormous adverse effects. In most of the cases, patients could be re-exposed to ifosfamide either by reducing dosage of ifosfamide or by preventive application of thiamine and/or methylene blue.

To reflect these results the manuscript was amended accordingly.

Reviewer 3 Report

The description and study design are good.

Definitions of nephropathy and encephalopathy should be more properly described.

I have the impression that the percentage of encephalopathy is high; mesna dosage and other information are needed.

How many days is Ifosfamide given in one cycle.

In Figure3 (j), are you talking about INN? or encephalopathy?

Author Response

The description and study design are good.

We thank the reviewer for his or her comments and the evaluation of our manuscript.

Definitions of nephropathy and encephalopathy should be more properly described.

We agree with the reviewer that the terms "impaired renal function" and “encephalopathy” were not adequately defined so far. Impaired renal function was assumed from a serum creatinine value greater than or equal to 1.2 mg/dl. Encephalopathy, often referred to as IIN (ifosfamide-induced neurotoxicity), was determined and classified according to the Common Terminology Criteria for Adverse Events (CTCAE) version 5.0 guidelines (grade 1: mild symptoms, grade 2: moderate symptoms, limiting instrumental ADL (Activities of Daily Living) such as preparing meals, shopping for groceries or clothes, using the telephone, managing money, etc., grade 3: limiting self-care ADL, such as bathing, dressing and undressing, feeding self, using the toilet, taking medications, not bedridden, grade 4: life-threatening consequences, urgent intervention indicated, grade 5: death). Symptoms of encephalopathy include confusion, somnolence, coma, hallucinations, blurred vision, psychotic behavior, extrapyramidal symptoms, urinary incontinence, and seizures. These definitions are now specified in the current version of the revised manuscript.

I have the impression that the percentage of encephalopathy is high; mesna dosage and other information are needed.

The occurrence of encephalopathy is reported in the literature with a frequency of 15 to 30% (24, 27, 31-33). In our study cohort IIN was observed in 28.5 % of sarcoma patients receiving ifosfamide, which is in line with previous reports. Primary prophylaxis involves Mesna at a dose of 1000 mg/m2 body surface area (BSA) during treatment with fosfamide (until 12 hours after completion of the last ifosfamide dose). In addition, olanzapine 5 mg was administered once per day for 7 days. In the case of IIN grade 2 according to CTCAE, ifosfamide infusion was stopped and methylene blue (3x 50 mg i. v.) combined with thiamine (200 mg i. v.) was administered. After development of IIN, methylene blue and thiamine was given as secondary prophylaxis in each subsequent cycle.

How many days is Ifosfamide given in one cycle.

In our study different protocols containing ifosfamide were used. In combination with doxorubicin, ifosfamide was given for 3 days per cycle in soft tissue sarcomas (STS) and for 2 days per cycle in bone sarcomas (BS) (EUROBOSS protocol). In combination with cisplatin, ifosfamide was given for 2 days per cycle, as well as in combination with vincristine. Within the EURAMOS protocol for BS, ifosfamide was given in combination with etoposide over 5 days per cycle. In patients who also received radiation, ifosfamide was applied over 2 days per cycle. These details are given now in given in Table 2.

In Figure 3 (j), are you talking about INN? or encephalopathy?

Figure 3 (j) illustrates overall survival (OS) of all sarcoma patients without occurrence of IIN listed in blue and with occurrence of IIN listed in red, whereby OS in the group of patients with IIN is worse. In the former version of the manuscript colors were changed. We apologize for this and corrected it in the recent version of the manuscript.

Reviewer 4 Report

The authors fail to recognize established factors related to ifosfamide neurotoxicity and have not included them in their model.  Albumin level (Wiltshaw) and acid-base balance (Rosen, Patel) have been long noted as prognostic factors.  The survival advantage noted without corresponding progression-free survival is likely due to known factors of disease status or histology, not accounted for in prognostic variable analysis.  The entire analysis needs redoing or simply abandoning the project.  This is a good example of statistics in the absence of logic leading to erroneous conclusions.

Author Response

The authors fail to recognize established factors related to ifosfamide neurotoxicity and have not included them in their model. Albumin level (Wiltshaw) and acid-base balance (Rosen, Patel) have been long noted as prognostic factors. 

We thank the reviewer for his or her insightful comments concerning IIN-related factors. Naturally, we are familiar with the studies the reviewer listed (Wiltshaw, Rosen and Patel) and the role of hypalbuminemia and acid-base balance in the development of encephalopathy. Here we present a retrospective single center analysis of risk factors for the development of IIN and the impact of IIN on PFS and OS in sarcoma patients. On the one hand the aim of this study was to evaluate additional factors that have influences the development of encephalopathy during ifosfamide therapy. Therefore, in addition to hypalbuminemia and acid-base status, further factors, such as neutrophil count, lymphocyte count, hemoglobin level, platelet count, level of serum creatinine, sodium, calcium, alanine aminotransferase (GPT), aspartate aminotransferase (GOT), gamma-glutamyl transferase (GGT) and C-reactive protein (CRP) were consecutively analyzed. Albumin level could retrospectively be determined in 31 out of 49 patients. Interestingly, with a mean of 3.45 g/dl (0.64 SD) patients with IIN should no significant alterations regarding to serum albumin level. Unfortunately, acid-base balance is not part of the standard laboratory for inpatient admission. Therefore, these values could not be determined in our study cohort. We regret this and will implement regular determinations of acid-base balance for future patients on ifosfamide therapy at our department.

The survival advantage noted without corresponding progression-free survival is likely due to known factors of disease status or histology, not accounted for in prognostic variable analysis. 

As shown in Figures 3 (i), (j) and (k) both, PFS and OS were significant reduced in patients presenting with IIN compared to STS patients without signs of IIN (median PFS 9 vs. 16 months, p < 0.001, Fig. 3 i; OS HR 2.04, 95 % CI: 1.2-3.9, p < 0,001, Fig. 3 j).

The entire analysis needs redoing or simply abandoning the project.  This is a good example of statistics in the absence of logic leading to erroneous conclusions.

In our study we not only confirm previous data with regard to already known risk factors for IIN in a cohort of 174 sarcoma patients. We additionally identified several new potential influencing factors influencing IIN. The observation that an inflammatory state is associated with an increased risk of IIN is a valuable information and can be used prospectively to further investigate relationship of inflammation and IIN. Moreover, the easily measured blood markers found to predict IIN in our study can be used for clinical decision making.

Round 2

Reviewer 2 Report

Thank you for taking my feedback into consideration and amending the manuscript accordingly. The only feedback I would like to add, is that it is  unclear in case of frequently measured laboratory parameters over the course of therapy, which value was selected in this study (e.g. at onset of INN, at initiation of therapy with ifosfamide? and what if the measurement was missing at that time point? Did you then choose a value at another time point?).

Furthermore, I feel that you cannot state that these laboratory parameters predict INN if these parameters were measured at the same time as onset of INN. It would only be predictive if it the parameters were measured before onset of INN. If the lab parameters were selected at onset of INN, I would suggest to use 'lab parameters are associated with INN' instead of lab parameters predict INN. 

Author Response

Thank you for taking my feedback into consideration and amending the manuscript accordingly. The only feedback I would like to add, is that it is  unclear in case of frequently measured laboratory parameters over the course of therapy, which value was selected in this study (e.g. at onset of INN, at initiation of therapy with ifosfamide? and what if the measurement was missing at that time point? Did you then choose a value at another time point?).

We thank the reviewer for bringing up this point. Due to the fact, that treatment with antiemetics or corticosteroids might influence the investigated laboratory parameters, laboratory values were analyzed at initiation of ifosfamide treatment. To investigate this we additionally analyzed level for Hb, CRP and ALC at treatment initiation and 48 hours later. As shown in Fig. for the reviewer 1 we observed relevant changes in CRP level and ALC count during treatment.

Of note, in all patients all relevant laboratory parameters were available at time point of treatment initiation.

Furthermore, I feel that you cannot state that these laboratory parameters predict INN if these parameters were measured at the same time as onset of INN. It would only be predictive if it the parameters were measured before onset of INN. If the lab parameters were selected at onset of INN, I would suggest to use 'lab parameters are associated with INN' instead of lab parameters predict INN.

We agree with the expert but since the laboratory parameters were analyzed at time point of treatment initiation we describe predictive parameters. We again apologize for not having stated this in the previous version of the manuscript. In order to take this into account we amended the manuscript accordingly.

Reviewer 4 Report

The authors have come up with an association between ifosfamide neurotoxicity, prognosis, and inflammatory surrogate parameters.  The relevance of the observation is uncertain, and it is unclear how this would modify the approach to therapy.  It would be interesting to perform the same analysis on patients who did not receive ifosfamide, either due to alternate therapy or in a relapse setting, to determine if these are prognostic factors for patients with sarcoma in general.

Author Response

The authors have come up with an association between ifosfamide neurotoxicity, prognosis, and inflammatory surrogate parameters.  The relevance of the observation is uncertain, and it is unclear how this would modify the approach to therapy.  It would be interesting to perform the same analysis on patients who did not receive ifosfamide, either due to alternate therapy or in a relapse setting, to determine if these are prognostic factors for patients with sarcoma in general.

We thank the reviewer for this bringing up this interesting point. In order to further investigate the prognostic role of inflammatory parameters identified in our cohort we analyzed the prognostic relevance of CRP, Calcium, GGT, LDH, ALC and Hb for PFS and OS in patients with and without IIN. Whereas CRP and ALC were shown to be significantly associated with PFS in patients with IIN, in patients without signs of IIN none of the identified factors were correlated with PFS (Suppl. Fig. 2.). Interestingly, CRP and Hb level predicted OS in patients with and without IIN suggesting, that these markers might be independently associated with a worse prognosis in sarcoma. The observation, that inflammation and especially CRP and Hb level are associated with worse outcome in sarcoma patients has been shown previously (Nakamura et al., Anticancer Res. 2017). To further validate this observation we investigated the prognostic role of ALC, HB, LDH and CRP in a small cohort of sarcoma patients without ifosfamide treatment (Suppl. Fig. 2, suppl Table 2). Of note, in this sarcoma patients Hb and CRP were shown to associated with a worse prognosis. We conclude that inflammation represents an independent risk factor in sarcoma patients associated with an higher risk for the development of IIN. However, the underlying mechanism explaining the relationship of IIN and the inflammatory state is unclear so far. Nevertheless, several studies have shown that permeability of the blood brain barrier is increased during systemic inflammation. As a result a rational link between inflammatory state and the risk of IIN might be speculated.

To take this into account, beside including Fig. 4, suppl Fig. 2 and suppl. Table 2 the manuscript was amended respectively.